# CTP: A Causal Interpretable Model for Non-Communicable Disease Progression Prediction

## Abstract

Non-communicable disease is the leading cause of death, emphasizing the need for accurate prediction of disease progression and informed clinical decision-making. Machine learning (ML) models have shown promise in this domain by capturing non-linear patterns within patient features. However, existing ML-based models cannot provide causal interpretable predictions and estimate treatment effects, limiting their decision-making perspective. In this study, we propose a novel model called causal trajectory prediction (CTP) to tackle the limitation. The CTP model combines trajectory prediction and causal discovery to enable accurate prediction of disease progression trajectories and uncover causal relationships between features. By incorporating a causal graph into the prediction process, CTP ensures that ancestor features are not influenced by the treatment of descendant features, thereby enhancing the interpretability of the model. By estimating the bounds of treatment effects, even in the presence of unmeasured confounders, the CTP provides valuable insights for clinical decision-making. We evaluate the performance of the CTP using simulated and real medical datasets. Experimental results demonstrate that our model achieves satisfactory performance, highlighting its potential to assist clinical decisions.

## 1 Introduction

Non-communicable disease (NCD), e.g., Alzheimer's disease and heart failure, is the leading cause of death across the world (Bennett et al., 2018). Prognosis prediction is regarded as a method for achieving precise medicine to NCDs (Gill, 2012). The assumption is that if we can predict prognosis (e.g., die in one year) accurately, we can adopt target treatment in advance for patients with bad prognosis, and the bad prognosis may be prevented. The more accurate the model, the more precise the clinical decision. Under the assumption, recent studies focus on utilizing machine learning (ML) methods to obtain more accurate prediction models (Coorey et al., 2022).

However, whether the assumption is right in NCDs is challenged by clinicians (Wilkinson et al., 2020). NCDs usually have heterogeneous etiologies. Different patients usually require different therapies. Nevertheless, current ML-based models just foreshow the outcome of a patient. No matter how accurate, they did not inform clinicians which therapy may be helpful, so assisting clinical decisions is infeasible. What clinicians require are causal interpretable models that not only predict prognosis accurately but also answer counterfactual inquiries such as "What will happen to the patient if we control the value of a feature from $a$ to $b$?" (Moraffah et al., 2020). Clinical decision-making support is feasible when the model can help clinicians anticipate the consequences of their actions (Coorey et al., 2022). Most ML models (even explainable ML models) cannot answer such inquiries as they only capture correlational relationships between features. Although there are studies focused on estimating treatment effect, their application perspective is restricted because they usually only investigate the effect of a drug on a predefined binary end-point (Yao et al., 2021).

This study aims to investigate a more general problem that predicts the disease progression trajectory of a patient when we control the value of a feature according to observational data (Figure 1 (a)). Compared to previous treatment effect studies, we regard a treatment can be direct, dynamic, and continuous control to an arbitrary feature and not necessarily the usage of a drug. We regard the outcomes are continuous trajectories of all interested features rather than a predefined event (Yao et al., 2021; Ashman et al., 2023). We need to tackle two problems in achieving this goal. First, we

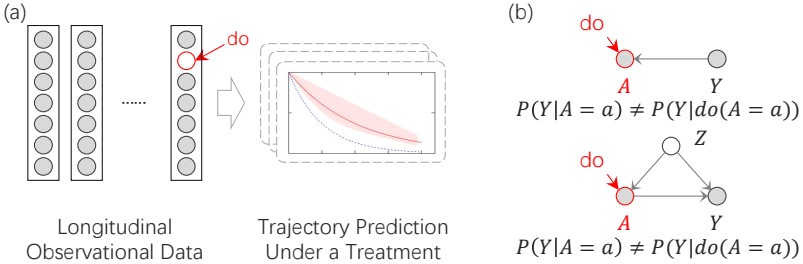

Figure 1: Goal and challenge. (a) This study investigates the progression of features under dynamic, continuous control of treatment via observational data. The blue line is the original trajectory while the red is the trajectory under a treatment, and the red region is the possible bound. (b) The main challenge is that the causal structure is unknown and there may exist unobserved confounders.

do not know the causal structure between features. A controlled feature $A$ may be a consequence of interested features $Y$ (i.e., $A \leftarrow Y$). If $A$ is a consequence of $Y$, our model is expected to generate unaffected trajectories of $Y$ when we control $A$ (Figure 1 (b)) (Neal, 2020). Second, as the development mechanisms of many NCDs are unclear, there may exist unmeasured confounders that we do not realize. Our models need to generate reliable estimations when a confounder exists (De Brouwer et al., 2022). Intuitively, our model needs to discover the causal structure between features from observational datasets and ensure the trajectory of every feature is predicted only by its historical value and its causative features. Then, we can generate unaffected trajectories of $Y$ under treatment $A$ when $A$ is a descendant of $Y$. In this study, we treat all observable features as $Y$. It's important to note that the effect of an unmeasured confounder may be unidentifiable as there may be infinite models that can generate the observed dataset when an unmeasured confounder exists (Gunsilius, 2021). Therefore, we additionally estimate the possible bounds of the treatment effect.

We designed a causal trajectory prediction (CTP) model consisting of two phases. The CTP predicts the progression trajectory of features in a causal interpretable manner in the first phase. It formulates the trajectory prediction problem as solving ordinary differential equations (ODE). It estimates features' derivatives using neural networks, and trajectories can be predicted via a numerical neural ODE solver (Chen et al., 2018). It also adopts a neural connectivity matrix to evaluate the predictive effect with respect to each feature pair (Lachapelle et al., 2020). To ensure each feature is predicted only by its historical value and its causative features, we applied a sparse penalty and a score-based penalty to the neural connectivity matrix. Previous studies have demonstrated that it is possible to discover causal structures in a linear dynamical system using penalties (Stanhope et al., 2014; Brunton et al., 2016; Chen et al., 2021). We extend this approach to a non-linear system in this study. Once the CTP model is optimized and the causal structure is identified, we retrain a group of independent CTP models to estimate the bounds of the treatment effect (the second phase). The training goal is that the group of new CTP models needs to fit the observed dataset accurately and generate trajectories as different as possible when we apply a treatment. Finally, we estimate the treatment effect bounds by analyzing trajectories generated by the group of retrained models.

We investigated the performance of the CTP model in discovering causal relationships between features, predicting feature progression trajectories, and predicting treatment effects. We utilized one real longitudinal medical dataset and four simulated datasets to evaluate model performance. Experiment results indicate that our framework is able to reconstruct the causal graph within features, obtain satisfactory predictive performance, and evaluate the bound of treatment effects well. Therefore, we believe the CTP model introduces an effective way to assist clinical decisions.

## 2 METHOD

### 2.1 PRELIMINARY

We denote a dataset consisting of a sequence of a two-element tuple $(s_i, l_i)_{i=1}^N$. $s_i = (v_{ij}, m_{ij}, t_{ij})_{j=1}^{N_i^s}$ indicates a sequence of visit data of a patient until time point $t_{iN_i^s}$, and $N_i^s$ means

the number of visits. $v_i$ denotes progression trajectory of the $i$-th patient, and $v_i(t)$ is a vector with $K$ elements that denotes patient characteristics at timepoint $t$. $v_{ij}$ means data in the $j$-th visit (or it can be regarded as the abbreviation of $v_i(t_{ij})$), and $v_{ij}^k$ denotes the value of the $k$-th feature. $v_{ij}^k$ can be a continuous variable or a discrete variable. In this study, we presume discrete variables are binary for simplicity, while the proposed method can be generalized to handle categorical variables naturally. $m_{ij} \in \{0,1\}^K$ indicates whether a corresponding feature in $v_{ij}$ is missing (1 indicates missing). $l_i = (v_{ij}, m_{ij}, t_{ij})_{j=N_i^s+1}^{N_i^s+N_i^l}$ indicates a sequence of (label) data need to be predicted.

We use a numerical adjacency matrix $D \in \mathbb{R}_{\geq 0}^{(K+1) \times (K+1)}$ to describe the casual structure between features (Bhattacharya et al., 2021). $D_{kl} = 0$ indicates the $k$-th feature is not the cause of the $l$-th feature, and $D_{kl} \neq 0$ means $k$-th feature is a cause of the $l$-th feature. Without loss of generality, $D$ uses an extra dimension to model the causal relationship between observed features and unmeasured confounders (Löwe et al., 2022). In this study, we presume the progression of the unmeasured confounder is not affected by any observed features.

## 2.2 CAUSAL TRAJECTORY PREDICTION

We estimate progression trajectories of features $\hat{v}_i(t)$ by solving ODEs (Chen et al., 2018). Our CTP model first estimates the value of patient characteristics $\hat{v}_{i0} \in \mathbb{R}^{K+1}$ at an initial time point $t_0$. We follow a standard variational autoencoder to estimate the posterior of the $q(\hat{v}_{i0}|s_i)$, where $q(\hat{v}_{i0}|s_i)$ follows a Gaussian distribution with a diagonal covariance matrix, $t_0$ is a custom number. We use a long-short-term memory (LSTM) network parameterized by $\phi$ to estimate $\mu_i, \sigma_i$ (Equation 1). Then, we randomly sample the $\hat{v}_{i0}$ via the reparameterization trick (Equation 2) (Kingma & Welling, 2014).

$$[\mu_i, \sigma_i] = \text{LSTM}([v_{ij}, m_{ij}, t_{ij}]_{j=1}^{N_i^s}; \phi). \tag{1}$$

$$q(\hat{v}_{i0}|s_i) = \mathcal{N}(\hat{v}_{i0}|\mu_i, \sigma_i). \tag{2}$$

$$\bar{v}_i^k(t) = \begin{cases} \hat{v}_i^k(t), & \text{continuous variable} \\ \text{Gumbel\_Sigmoid}(\hat{v}_i^k(t)), & \text{discrete variable} \end{cases} \tag{3}$$

$$\frac{d\hat{v}_i^k(t)}{dt} = f_{\theta_k}(\bar{v}_i(t) \circ \mathcal{M}_k). \tag{4}$$

$$\hat{v}_i(t) = \text{ODESolver}(f_\theta, \hat{v}_{i0}, t_0, t). \tag{5}$$

We first map the $\hat{v}_i^k(t)$ to $\bar{v}_i^k(t)$ (Equation 3), which is an identical mapping for continuous variables and discretizes logit for discrete variables (Jang et al., 2017). Then, we use $K+1$ independent components $f_{\theta_k}$ to predict each feature derivatives $\frac{d\hat{v}_i^k(t)}{dt}$ (Equation 4), where $\circ$ means element-wise multiplication, and each $f_{\theta_k}$ is a feed-forward network (FFN). $\mathcal{M}_k$ is a causal mask which will be introduced later and it can be regarded as a vector whose all elements are one at present. The $\hat{v}_i(t)$ can be estimated according to $f_\theta$ and $v_{i0}$ via a numerical ODE solver (Equation 5) (Chen et al., 2018). As ODE-based models are only capable of modeling the dynamics of continuous variables, the $\hat{v}_i^k$ represents logit values, rather than the true values, for discrete variables.

The core of causal interpretability is to ensure a feature is predicted only by itself and its cause features. Here, we introduce the **_neural connectivity matrix_** to evaluate the predicted effect of a feature pair (Lachapelle et al., 2020). Specifically, the form of an FFN (without bias term, the output is a real number) follows $o = W_N \sigma(\cdots \sigma(W_2(\sigma(W_1 x))))$. $x \in \mathbb{R}^{K+1}$ is the input, $o \in \mathbb{R}$ is the output, $\sigma$ is the non-linear activation, and $W_1, \cdots, W_N$ is a series of weight matrices (vectors). The connectivity vector $C \in \mathbb{R}^{K+1}$ follows:

$$C = |W_N| \cdots |W_2||W_1|. \tag{6}$$

It is easy to find that $C_j = 0$ indicates the $j$-th input element does not affect the output. We can derive $\widetilde{D}_{jk} = C_j^k$, where $C^k$ is the connectivity vector for $f_{\theta_k}$. $\widetilde{D}$ analogs an adjacent matrix where

$\widetilde{D}_{jk} \neq 0$ represents the $j$-th feature can predict the $k$-th feature somehow. We can use sparse penalty $g(\theta) = \sum_{i=1}^{K+1} \sum_{j=1}^{K+1} (\widetilde{D})_{ij}$ to remove spurious causal connections (Brunton et al., 2016).

Moreover, the causal relationship between features in many diseases can be characterized as a directed acyclic graph (DAG) and there is no feedback (Blaser et al., 2015; Suttorp et al., 2015). For example, the casual relation of features in the amyloid beta pathway in the progression of Alzheimer's disease formulates a DAG (Hao et al., 2022). Therefore, we additionally presume the causal graph is a DAG and applied an extra score-based constraint to $\widetilde{D}$ in this study (Equation 7).

$$h(\theta) = \text{Tr}(\exp((1-I) \circ \widetilde{D})) - K, \tag{7}$$

where Tr means the trace of a matrix, and $\exp(A)$ denotes the exponential of a non-negative square adjacent matrix $A$ that is defined as the infinite Taylor series, i.e., $\exp(A) = \sum_{k=0}^{\infty} \frac{1}{k!} A^k$, $A^0 = I$. We use $(1-I)$ to denote we allow self-loop (i.e., a feature is able to predict its derivative). Zheng et al. (2018) proofed $A_{ij}^k$ indicates a weighted path count from element $i$ to $j$ after $k$ steps, and the count is a non-negative number. If the $A$ represents a cyclic graph, there must be some $A_{ii}^k > 0$, and cause $\text{Tr}(\exp(A)) - K > 0$. The score-based DAG constraint equals zero if and only if the $(1-I) \circ \widetilde{D}$ represents a DAG. Once the constrained hold, it is possible that every $v_i^k(t)$ is predicted only by itself, and its direct cause features.

We optimize parameters by minimizing the $\mathcal{L}$ (Equation 8). The optimizing goal is perfectly reconstructing observed data when $j$ is less or equal to $N_i^s$, and accurately predicting future trajectory when $j$ is greater than $N_i^s$. The mean square error (MSE) is used to measure the difference between the predicted value and true value in continuous variables, and cross-entropy (CE) is used for discrete variables. $B$ is a mini batch of dataset and $|B|$ indicates its size.

$$\mathcal{L} = \sum_{i,j,k}^{|B|,N_i^s+N_i^l,K} \begin{cases} \text{MSE}(v_{ij}^k, \hat{v}_{ij}^k), & v_{ij}^k \text{ is continuous} \\ \text{CE}(v_{ij}^k, \hat{v}_{ij}^k), & v_{ij}^k \text{ is discrete} \\ 0, & v_{ij}^k \text{ is missing} \end{cases} . \tag{8}$$

Finally, the objective function of this study follows Equation 9.

$$\min_{\theta, \phi} (\mathcal{L} + \beta g(\theta)) \quad \text{s.t. } h(\theta) = 0, \tag{9}$$

where $\beta$ is the weight of the sparse penalty. The augmented Lagrangian method can optimize parameters by solving a sequence of unconstrained subproblems (Lachapelle et al., 2020; Zheng et al., 2018). In our study, each subproblem is:

$$\mathcal{L}_{final} = \mathcal{L} + \beta g(\theta) + \frac{\rho}{2} h(\theta)^2 + \alpha h(\theta), \tag{10}$$

where $\rho$, $\alpha$ are penalty weights, respectively. We approximately solve each subproblem via the stochastic gradient descent method (details in Appendix C.1). We adopted the adjoint sensitive method to make the ODE solver differentiable (Chen et al., 2018).

## 2.3 CAUSAL GRAPH IDENTIFICATION

Our CTP model still faces challenges in discovering causal relationships between features. The first challenge is that the value of $\widetilde{D}_{ij}$ cannot be penalized to exactly zero, but a very small number, as we use a numerical optimizer. The model is also fragile because the neural ODE and matrix exponential operation in the CTP model is sensitive to parameter initialization and input noise (Rodriguez et al., 2022). Even if the CTP model successfully converges and obtains good prediction performance, it is easy to identify causal edges with the wrong causal direction. We adopted an iterative algorithm to tackle the above limitations. The algorithm uses $\mathcal{M} \in \{0,1\}^{(K+1) \times (K+1)}$ to describe causal relation between features and $\widetilde{\mathcal{M}} \in \{0,1\}^{(K+1) \times (K+1)}$ to determine whether the causal relationship is certain. We use the Equation 11 to initialize $\mathcal{M}$ and $\widetilde{\mathcal{M}}$, where $\mathcal{M}_{ij} = 1$ indicates $i$-th feature is the cause of the $j$-th feature, and $\widetilde{\mathcal{M}}_{ij} = 1$ indicates the $\mathcal{M}_{ij}$ is not certain. In the beginning, most

causal relations are uncertain. Of note, we set $\mathcal{M}_{ij} = 0$ when $i = K + 1$ and $j \leq K$ because we presume the hidden confounder is not the consequence of observed features.

$$\mathcal{M}_{ij}, \widetilde{\mathcal{M}}_{ij} = \begin{cases} 1, \ j \leq K \\ 0, \ j = K + 1 \text{ and } i \leq K \\ 1, \ j = K + 1 \text{ and } i = K + 1 \end{cases}. \tag{11}$$

The algorithm first repeatedly optimizes independent CTP models until $N$ models converge successfully. We presume these models are more probable to identify correct causal relationships. Then, we analyze the neural connectivity matrix of each model. We treat elements in neural connectivity matrix larger than a threshold is valid (Lachapelle et al., 2020). We use the $e_{ij}/N$ to determine how many models treat the connection $i \to j$ is valid, where $e_{ij}$ the number of models that regard $i \to j$ is valid. If the value is larger than an accept ratio $\rho$, we will treat the connection is certainly valid and set $\mathcal{M}_{ij} = 1$ and $\widetilde{\mathcal{M}}_{ij} = 0$. $e_{ij}/N$ is less than $1 - \rho$ indicates the connection is certainly invalid and we we set $\mathcal{M}_{ij} = 0$ and $\widetilde{\mathcal{M}}_{ij} = 0$. We repeat the process until all casual relations become certain (i.e., $\sum_{ij}(\widetilde{\mathcal{M}}_{ij}) = 0$). The pseudocode of the algorithm is described in Appendix C.2.

## 2.4 TREATMENT EFFECT PREDICTION

We conduct treatment effect prediction under two assumptions. (1) The optimized CTP model $M^\star$ predicts the feature progression trajectory accurately. (2) The $M^\star$ summarizes reliable causal structure between observed features. However, it is challenging to evaluate the effect of a treatment when an unmeasured confounder exists because parameters in the $M^\star$ may be not the only solution to the prediction problem (Miao et al., 2011). There maybe infinite other parameters that can generate the observed dataset, and these choices of parameters generate different trajectories under a treatment. In this study, we presume all these parameters are located in a connected region.

As we lack the ability to identify which choice of parameters is better, this study estimates the feature trajectories and probable bounds under a treatment, rather than deconfounding (Cao et al., 2023). We adopt an intuitive idea that retrains a series of new independent CTP models that cover the region of the parameter. Generally, the new group of retrained CTP models share the same structure and parameters according to the $M^\star$. We use $\text{do}(A^{t_a} = a)$ to denote a treatment, which means fixing the value of $\bar{v}^A$ to $a$ from a time point $t_a$, regardless of its original value. Given patient data $s_i$ and a new model $M^l$, we can predict trajectories of other features $^l\hat{v}_i^k(t)$ under the treatment $\text{do}(A^{t_a} = a)$, where $l$ is the index of a new CTP model. Of note, $M^l$ infers the value of unmeasured confounders so that we evaluate the effect of confounders. We record the patient characteristics under a treatment $^l\hat{v}_i^k(t_o)$ at a randomly selected time point $t_o$ after treatment. To make trajectories of $M^l$ as dissimilar as possible, we maximize the pair-wise distance between recorded $^l\hat{v}_i(t_o)$ (Equation 12). In this study, we applied the simplest p-norm distance for computational efficiency. However, a more sophisticated loss function such as Wasserstein distance is also applicable (Balazadeh Meresht et al., 2022). The optimization problem is a min-max problem that minimizes the prediction loss $\mathcal{L}_p$ (Equation 13) and maximizes the treatment $\mathcal{L}_t$. We use two optimizers to update parameters alternatively (details in Appendix C.3), which is widely used in similar studies (Kostrikov et al., 2020). Once the series of new CTP models are retrained, we treat contours (i.e., $\max(^l\hat{v}_i(t))/\min(^l\hat{v}_i(t))$) as trajectory bounds. We use the expectation of trajectories of retrained models as the trajectories under the treatment.

$$\mathcal{L}_t = \sum_{i=1}^{|B|} \sum_{j=1}^{L-1} \sum_{k=j+1}^{L} \text{Distance}(^j\hat{v}_i(t_o), {}^k\hat{v}_i(t_o)), \tag{12}$$

$$\mathcal{L}_p = \sum_{l,i,k}^{L,|B|,K} \begin{cases} \text{MSE}(v_i^k(t_a), {}^l\hat{v}_i^k(t_a)), \ v_i^k(t_a) \text{ is continuous} \\ \text{CE}(v_i^k(t_a), {}^l\hat{v}_i^k(t_a)), \ v_i^k(t_a) \text{ is discrete} \end{cases}. \tag{13}$$

| | Hao | Zheng | MM-25 | MM-50 | ADNI |
|---|---|---|---|---|---|
| # of Samples | 1,024 | 1,024 | 1,024 | 1,024 | 275 |
| Avg. Visit | 15 | 15 | 15 | 15 | 3.7 |
| # Features | 4 | 4 | 20 | 45 | 88 |
| Avg. Interval | 1.00 | 2.00 | 0.25 | 0.25 | 1.65 |
| All Continuous | Yes | Yes | Yes | Yes | No |

Table 1: Dataset Statistics

## 3 EXPERIMENT

### 3.1 EXPERIMENT SETTINGS

**Dataset**. We used one real medical dataset and four simulated data to evaluate the performance of CTP, whose statistics are in Table 1. *Real Dataset*: ADNI dataset consists of a comprehensive collection of multi-modal data from a large cohort of subjects, including healthy controls, individuals with mild cognitive impairment, and patients with Alzheimer's disease (Petersen et al., 2010). The preprocessed ADNI dataset reserved 88 features (23 continuous and 65 discrete) of patients whose available visit record is equal to or greater than three. *Simulated Dataset*: Hao dataset records the progression of four features (amyloid beta ($A_\beta$), phosphorylated tau protein ($\tau_p$), neurodegeneration ($N$), and cognitive decline score ($C$)) of late mild cognitive impairment patients (Hao et al., 2022). Zheng Dataset: it records the progression trajectories of four features (i.e., $A_\beta$, tau protein $\tau$, $N$, and $C$) of Alzheimer's disease patients (Zheng et al., 2022). MM-25 and MM-50 datasets: we also generated two Michaelis-Menten (MM) kinetics datasets to evaluate the model in a high-dimensional scenario (Zhang et al., 2022). The MM-25 contains 20 features and the MM-50 contains 45 features. The Zheng dataset is a confounder-free dataset and the other three datasets contain unobservable confounders. All datasets were normalized before training. More detailed data generation and the preprocessing process are described in Appendix B.

**Baselines**. We used two commonly seen models and three recently proposed models as baselines. *LODE*: the Linear ODE baseline uses the same structure compared to the CTP, while it uses a linear function to model the derivatives of features. The LODE used the ridge loss to remove spurious connections (Brunton et al., 2016). *NODE*. Neural ODE also uses the same structure compared to the CTP, while it does not use ridge loss and DAG loss to optimize parameters (Chen et al., 2018). *NGM*: NGM uses the same structure compared to our CTP, while it only adds group ridge loss to the first layer of neural network to extract causality (Bellot et al., 2022). *TE-CDE*: TE-CDE adopts controlled differential equations to evaluate patient trajectory at any time point and uses an adversarial training approach to adjust unmeasured confounding (Seedat et al., 2022). *CF-ODE*: CF-ODE adopts the Bayesian framework to predict the impact of treatment continuously over time using NODE equipped with uncertainty estimates (De Brouwer et al., 2022).

**Treatment Settings**. We only conducted treatment effect analysis on four simulated datasets because it is impossible to access the counterfactual result of the ADNI dataset. We run 16 independent models for each dataset. *Hao Dataset*: we set the neurodegenerative value (i.e., $n$) to zero at time point 52. Then, we observed the feature progression trajectories under the treatment from 52 to 60. *Zheng Dataset*: we set the neurodegenerative value (i.e., $n$) to zero at the DPS time zero. Then, we observed the feature progression trajectories under the treatment from 0 to 20. *MM-25 and MM-50 Dataset*. We set the value of the No. 10 node to one at the time point one. We recorded the feature progression trajectories under the treatment from 1 to 10.

**Metrics**. *Trajectory Prediction*: We used MSE to evaluate the prediction performance on continuous features and the macro average area under the receiver operating curve (AUC) to evaluate the prediction performance on discrete features. *Causal Discovery*: We investigated the causal discovery performance of the CTP and baselines by analyzing the neural connectivity matrix $\widetilde{D}$. We regard a casual edge is inexistent if its corresponding element is less than 0.0001, and vice versa. Then, we use accuracy, F1, and AUC to evaluate causal discovery performance. *Treatment Effect Prediction*: We used the MSE between the true value and estimated trajectories to evaluate the treatment effect prediction performance.

**Extended Experiments**. We released extended experiment results in Appendices.

| Model | Dataset | Reconstruct | | Predict | |
|---|---|---|---|---|---|
| | | MSE | AUC | MSE | AUC |
| LODE | ADNI | 0.26±0.02 | 0.67±0.02 | **0.26±0.02** | 0.46±0.02 |
| NODE | ADNI | 0.22±0.01 | 0.69±0.01 | 0.45±0.03 | 0.54±0.02 |
| NGM | ADNI | 0.23±0.02 | 0.68±0.01 | 0.34±0.02 | 0.53±0.01 |
| TE-CDE | ADNI | 0.24±0.01 | 0.66±0.01 | 0.30±0.01 | 0.52±0.01 |
| CF-ODE | ADNI | 0.21±0.03 | 0.69±0.02 | 0.32±0.03 | 0.54±0.01 |
| CTP | ADNI | **0.20±0.01** | **0.73±0.01** | 0.29±0.02 | **0.55±0.02** |

Table 2: ADNI Data Prediction Performance

| Model | Hao Dataset | | | Zheng Dataset | | |
|---|---|---|---|---|---|---|
| | ACC. | F1 | AUC | ACC. | F1 | AUC |
| LODE | 0.53±0.01 | 0.60±0.01 | 0.67±0.01 | 0.58±0.01 | 0.64±0.02 | 0.64±0.03 |
| NODE | 0.50±0.02 | 0.57±0.03 | 0.54±0.10 | 0.49±0.01 | 0.62±0.02 | 0.53±0.01 |
| NGM | 0.54±0.02 | 0.60±0.02 | 0.71±0.05 | 0.59±0.02 | 0.65±0.02 | 0.61±0.01 |
| TE-CDE | 0.53±0.02 | 0.59±0.03 | 0.57±0.10 | 0.56±0.03 | 0.65±0.03 | 0.54±0.02 |
| CF-ODE | 0.54±0.02 | 0.58±0.03 | 0.55±0.10 | 0.57±0.01 | 0.65±0.02 | 0.55±0.01 |
| CTP | **0.56±0.01** | **0.62±0.02** | **0.81±0.03** | **0.61±0.01** | **0.67±0.01** | **0.66±0.00** |
| CTP* | 1.00 | 1.00 | / | 0.88 | 0.93 | / |
| | MM-25 Dataset | | | MM-50 Dataset | | |
| | ACC. | F1 | AUC | ACC. | F1 | AUC |
| LODE | 0.75±0.01 | 0.76±0.01 | 0.82±0.01 | 0.87±0.01 | 0.87±0.01 | 0.89±0.01 |
| NODE | 0.51±0.02 | 0.67±0.03 | 0.53±0.10 | 0.50±0.02 | 0.66±0.01 | 0.57±0.02 |
| NGM | 0.80±0.02 | 0.81±0.02 | 0.67±0.05 | 0.87±0.02 | 0.87±0.01 | 0.66±0.01 |
| TE-CDE | 0.80±0.02 | 0.81±0.02 | 0.56±0.05 | 0.87±0.01 | 0.87±0.01 | 0.58±0.02 |
| CF-ODE | 0.53±0.01 | 0.56±0.02 | 0.53±0.03 | 0.87±0.03 | 0.87±0.02 | 0.58±0.02 |
| CTP | **0.82±0.01** | **0.83±0.02** | **0.88±0.03** | **0.89±0.01** | **0.89±0.01** | **0.90±0.01** |
| CTP* | 0.85 | 0.88 | / | 0.93 | 0.92 | / |

Table 3: Causal Discovery Performance. "CTP" and "CTP*" indicate the causal discovery performance of a CTP model without/with using the causal identification algorithm.

## 3.2 TRAJECTORY PREDICTION

We investigated the disease progression trajectory prediction performance of the CTP model and baselines. Table 2 indicates that our CTP models obtained comparable performance to baselines in the ADNI dataset. It obtained the first place in reconstruct MSE (0.20), reconstruct AUC (0.73), and predict AUC (0.55). The CTP model obtained second place in predict MSE (0.29), which is worse than the LODE model (0.26). Meanwhile, the experiment results on four simulated datasets showed our CTP model also obtained better or comparable performance compared to baselines, whose detail is in Appendix E.1. Although the goal of this study is not to propose a more accurate predictive model, these experiment results demonstrated our CTP model is able to obtain state-of-the-art (SOTA) or nearly SOTA performance in prognosis prediction tasks.

## 3.3 CAUSAL DISCOVERY

We only investigated causal discovery performance on four simulated datasets because we cannot access the true causal graph of the ADNI dataset (Table 3). It is not surprising that the NODE model cannot extract causal relations from features, as its AUC is less than 0.57 in all four datasets. The TE-ODE and CF-ODE also obtained unsatisfactory performance as they are not designed to extract causal relations between features. They require prior causal information to estimate the treatment effect. The LODE and NGM achieve significantly better performance benefits from utilizing ridge loss. The NGM outperforms the LODE, which may be attributed to the usage of neural networks. Furthermore, We find that our CTP model can identify causal relations between features better than all baselines, and its performance can be further improved by utilizing the causal identification algorithm. For example, the original CTP model only obtained 0.56 causal discovery accuracy in the Hao dataset. However, its causal discovery performance can be significantly improved if we apply the causal identification algorithm. The CTP* model obtained 0.44, 0.27, 0.03, 0.04 performance gains in accuracy concerning four datasets and 0.38, 0.26, 0.05, 0.03 performance gains in F1.

| Model | Hao Dataset | | | Zheng Dataset | | |
|---|---|---|---|---|---|---|
| | Full | Near | Far | Full | Near | Far |
| LODE | 0.77±0.08 | 0.33±0.05 | 1.37±0.14 | 0.79±0.05 | 0.67±0.02 | 1.05±0.10 |
| NODE | 1.08±0.13 | 0.54±0.08 | 1.85±0.19 | 2.32±0.01 | 1.39±0.00 | 4.56±0.02 |
| NGM | 0.25±0.01 | 0.25±0.01 | 0.26±0.01 | 0.78±0.04 | 0.66±0.03 | 1.00±0.03 |
| TE-CDE | 0.32±0.02 | 0.32±0.01 | 0.31±0.01 | 3.88±0.01 | 1.75±0.01 | 9.21±0.01 |
| CF-ODE | 0.57±0.03 | 0.38±0.02 | 0.84±0.05 | 0.36±0.03 | **0.24±0.02** | 0.55±0.06 |
| CTP | **0.16±0.01** | **0.16±0.00** | **0.16±0.01** | **0.32±0.03** | 0.26±0.02 | **0.47±0.03** |
| CTP$^\star$ | 0.13±0.01 | 0.14±0.00 | 0.13±0.01 | 0.29±0.03 | 0.25±0.02 | 0.46±0.03 |
| | MM-25 Dataset | | | MM-50 Dataset | | |
| | Full | Near | Far | Full | Near | Far |
| LODE | 1.13±0.05 | 1.11±0.07 | 1.22±0.04 | 1.51±0.01 | 1.52±0.00 | 1.51±0.01 |
| NODE | 1.25±0.04 | 1.23±0.05 | 1.60±0.03 | 1.48±0.03 | 1.43±0.03 | 1.67±0.02 |
| NGM | 0.89±0.03 | 0.91±0.04 | **0.79±0.02** | 2.41±0.08 | 2.24±0.05 | 3.12±0.13 |
| TE-CDE | 1.25±0.09 | 1.19±0.11 | 1.52±0.15 | 1.50±0.04 | 1.45±0.05 | 1.70±0.08 |
| CF-ODE | 1.29±0.04 | 1.22±0.03 | 1.59±0.05 | 1.64±0.09 | 1.58±0.07 | 1.88±0.13 |
| CTP | **0.78±0.03** | **0.74±0.05** | 0.88±0.07 | **1.20±0.06** | **1.15±0.08** | **1.43±0.05** |
| CTP$^\star$ | 0.75±0.02 | 0.73±0.04 | 0.80±0.05 | 1.17±0.06 | 1.11±0.05 | 1.49±0.06 |

Table 4: Treatment Effect Prediction. The "Full" column indicates the general difference between predicted trajectories and oracle trajectories of features. The "Near" column indicates the differences in the trajectories before treatment and the first half observations after the treatment. The "Far" column indicates the differences in the second half observations after the treatment.

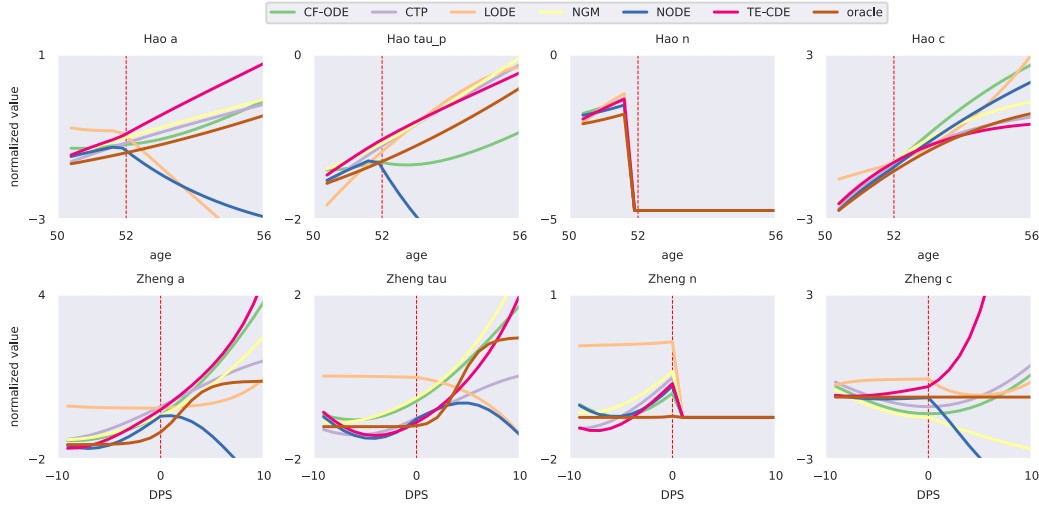

Figure 2: Treatment Effect Analysis (average trajectories of retrained CTP models and baselines).

## 3.4 TREATMENT EFFECT PREDICTION

Table 4 describes the performance of predicting progression trajectories under a given treatment of the CTP model and baselines in four datasets. The CTP (without utilizing a causal identification algorithm) obtained significantly better performance than all baselines. For example, The full MSE of NODE in the Hao dataset is 1.08, about six times more than the CTP model (0.16), and the full MSE of NODE is 1.39, about four times more than the CTP model (0.25). The TE-CDE and CF-ODE also obtained poor performance, which may be attributed to they only focus on deconfounding when prior causal information is available. The NGM model obtained the best performance among baselines, while the CTP model obtained better performance than the NGM. The performance of our CTP model is significantly better in the MM-25 and MM-50 datasets. These experiment results demonstrate our CTP model has good scalability. We also find that the CTP$^\star$ model obtained better performance by utilizing the causal identification algorithm, though the improvement is not significant.

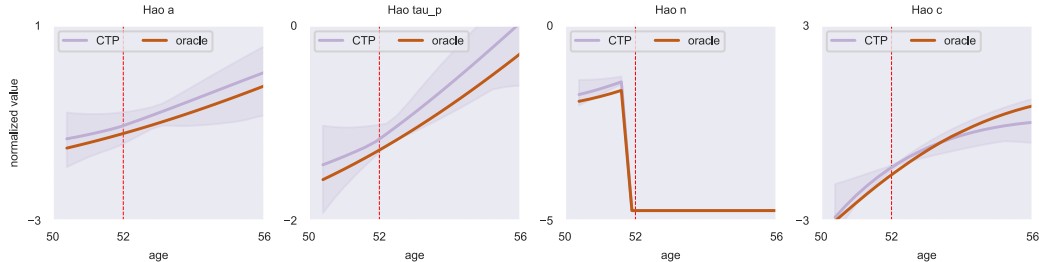

Figure 3: Trajectories Bounds under a Treatment of Hao Dataset

We qualitatively analyzed why our CTP model obtained better performance than baselines in Figure 2. For the space limit, we only draw trajectories of two randomly selected samples from the Hao dataset and Zheng dataset. The figure shows that some trajectories generated by baselines changed mistakenly after the treatment. For example, the predicted trajectory of $a$ in the Hao dataset of LODE and NODE decreases rapidly after we apply the treatment, while the treatment does not affect the trajectory of $a$ because $a$ is an ancestor feature of the $n$. Similar deviations also occur in other features. As these baselines utilized correlational relations, the treatment action brings unexpectable influence to the predicted feature. Although the LODE and the NGM model applied ridge loss to remove spurious connections, experimental results demonstrated they could not recover causal relations between features well. The incorporation of score-based DAG loss helps the model predict treatment effects better when there is no feedback between features.

We plot the bound of trajectories of a randomly selected sample to evaluate the bound qualitatively. The sample is from the Hao dataset as it contains an unobserved confounder (Figure 3). We find bounds generated by a group of retrained CTP models include true trajectories and are not very loose, indicating the bound may be helpful in clinical decision-making. We also describe the prediction performance of the retrained CTP models (Table 5) to investi-

| Dataset | Reconstruct | | Prediction | |
|---------|-------------|-----------|------------|-----------|
|         | Origin      | Retrained | Origin     | Retrained |
| Hao     | **0.01**    | **0.01**  | **0.02**   | 0.03      |
| Zheng   | 0.09        | **0.07**  | **0.09**   | **0.09**  |
| MM-25   | 0.04        | **0.03**  | 0.24       | **0.23**  |
| MM-50   | 0.04        | **0.04**  | **0.28**   | 0.29      |

Table 5: Prediction Performance of Retrained Models (Avg. MSE)

gate whether the retraining process deteriorates the trajectory prediction performance of models. We find that the retrained group of CTP models, though have different parameters, obtained the same or better performance in reconstructing input in the four datasets. They also obtained comparable performance to the original CTP model in predicting tasks. We may summarize that the retraining process not only helps us find the bound of treatment effect but also does not affect the prediction performance of the CTP model. The finding may be indirect evidence that there may be multiple models that can generate the observed dataset when an unobserved confounder exists.

## 4 CONCLUSION

We proposed a causal interpretable model that combines trajectory prediction and causal graph discovery to predict feature progression trajectories. The model ensures that each feature is predicted only by itself and its causal ancestors and tackles the issue of unmeasured confounders by identifying correlated errors and constraining the possible effect space of confounders using observed data. Experimental results demonstrate that the CTP model performs comparably or better than baselines in trajectory prediction. It obtained significantly better performance in predicting feature trajectories under a treatment. This model offers a novel approach to support clinical decision-making. In the future study, we will try to evaluate the causal discovery performance and treatment effect prediction performance of the CTP model in real medical datasets.

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

APPENDIX

# A RELATED WORK

## A.1 TRAJECTORY PREDICTION

Recurrent neural networks (RNN) and Transformer are widely used to learn representations from sequential patient data and predict their feature progression trajectories (Alaa & van der Schaar, 2019; Lim & van der Schaar, 2018). However, these two types of methods are designed to model discrete trajectories with fixed time intervals, while we may need to generate continuous patient trajectories. Recent studies usually adopt two kinds of methods to model continuous trajectories. The first method is to model the trajectory prediction problem as solving a dynamical system. Benefiting from the neural ODE solver (Chen et al., 2018), it is possible to optimize a neural network parameterized dynamical system, so that predicting continuous patient trajectory is feasible. For example, (Seedat et al., 2022) interprets the data as samples from an underlying continuous-time process and models its latent trajectory explicitly using controlled differential equations. (Alaa et al., 2022) temporally integrates embeddings of clinical codes and neural ODEs to learn and predict patient trajectories in electronic health records. The second method models continuous trajectories by revising the architecture of the RNN or transformer. For example, (Duan et al., 2019) utilizes the Hawkes process to learn the effect of time intervals and incorporate the Hawkes process into an RNN model. (Tan et al., 2020) proposes an end-to-end model that preserves the informative varying intervals by introducing a time-aware structure to directly adjust the influence of the previous status in coordination with the elapsed time. Besides, (Tipirneni & Reddy, 2022) regards the irregularity as data missing and overcomes the pitfall by treating time series as a set of observation triplets instead of using the standard dense matrix representation.

Although these models claimed they obtained impressive performance, they usually cannot be applied to treatment effect analysis due to they did not capture casual information within data.

## A.2 CAUSAL DISCOVERY

As far as we know, there are generally two kinds of methods that can discover casualties from sequential data. The first is to use an ODE-based linear dynamic system to model the data-generating process and adopt a sparse penalty (e.g., ridge loss) to eliminate unnecessary feature interactions. Previous studies have proved this approach can reconstruct causal structure correctly given observational data (Wang et al., 2022a). Known as physics informed network (PIN), this method is widely used to discover governing equations in physical processes (Brunton et al., 2016; Chen et al., 2021). The second method utilizes the Granger causality assumption that assumes each sampled variable is affected only by earlier observations. Then they summarize the causal graph by analyzing the Granger causality information. For example, (Cheng et al., 2023) use a threshold-based method to reserve probable Granger causality between features. (Amornbunchornvej et al., 2021) propose a regression and sparse-based algorithm to identify Granger causality from data. In this study, we adopt the first framework to discover a causal graph in a non-linear dynamic system.

## A.3 TREATMENT EFFECT PREDICTION WITH HIDDEN CONFOUNDER

Most previous studies tried to infer the value of the posterior distribution of hidden confounders given the observed data. Once the value or the distribution of the confounder is inferred, the treatment effect can be evaluated by adjusting. For example, (Bica et al., 2020) uses an RNN with multitask output to infer unmeasured confounders that render the assigned treatments conditionally independent. Similarly, (Ma et al., 2021) aims to learn how hidden confounders' representation over time by using observation data and historical information. (Cao et al., 2023) leverages Lipschitz regularization and neural-controlled differential equations to capture the dynamics of hidden confounders. (Ashman et al., 2023) develops a gradient and generative-based model to infer the existence and effect of hidden confounders simultaneously. (Wang et al., 2022b) proposes a clipping strategy to the inverse propensity score estimation to reduce the variance of the learning objective.

Although these studies obtained impressive progress, they did not discuss whether the effect of the unmeasured confounder is identifiable. In fact, the effect of unmeasured confounder may be generally

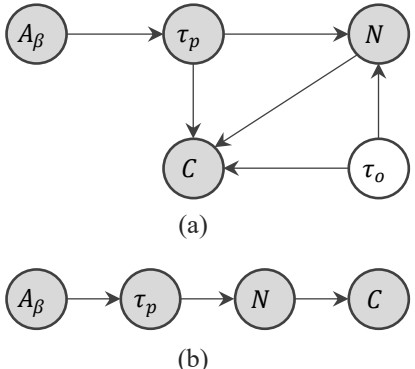

Figure 4: Causal Graph of simulated amyloid beta pathway data. (a) Hao model. (b) Zheng model. The causal graph only describes the causal relation between different features.

unidentifiable for both static dataset and sequential datasets (Gunsilius, 2019; 2021; Stanhope et al., 2014). Therefore, we argue it may be too optimistic to presume deconfounding is feasible and also try to estimate the bound of the treatment effect when a confounder exists (Balazadeh Meresht et al., 2022).

## B    DATA PAREPARING

### B.1    ADNI

The Alzheimer's Disease Neuroimaging Initiative (ADNI) is a multicenter, prospective, naturalistic dataset (Petersen et al., 2010). The ADNI database consists of a comprehensive collection of neuroimaging, clinical, and genetic data from a large cohort of subjects, including healthy controls, individuals with mild cognitive impairment, and patients with Alzheimer's disease. We have traversed the ADNI database, excluding mass spectrometry analysis, to match participant demographic information, biomarkers, and family cognitive status data.

We only reserved patients whose available visit record is equal to or greater than three to ensure patient data is longitudinal. Features whose missing rate is larger than 30% were discarded. Finally, medical records from 275 patients whose ages range from 55 to 90 years old were included. These patients contain 1,018 admission records and the average admission time is 3.7. The average interval between two visits is 1.65 years. Each admission record contains 88 features, where 23 features are continuous and 65 features are discrete. The continuous features consist of important biomarkers, e.g., tau protein, amyloid beta protein, and cognitive assessment results, e.g., Mini-Mental State Examination (MMSE). Discrete features recorded the usage of medicines.

### B.2    AMYLOID BETA PATHWAY SIMULATED DATASET

We used the amyloid beta pathway, an extensively investigated disease progression pathway of Alzheimer's disease, to conduct analysis (Hao et al., 2022; Zheng et al., 2022; Mielke et al., 2017). We used two recently published ODE-based bio-mathematical models to instantiate the pathway.

**Hao model** (Hao et al., 2022). This model includes five features, i.e., amyloid beta ($A_\beta$), phosphorylated tau protein ($\tau_p$), nonamyloid-dependent tauopathy ($\tau_o$), neurodegeneration ($N$), and cognitive decline score ($C$). It describes the dynamics of the pathway on three different groups of patients, i.e., Alzheimer's disease, late mild cognitive impairment (LMCI), and cognitive normal groups. The dynamics of the three groups share the same form of ODEs but with different parameters (Equation 14). In this study, we chose the group of parameters and initial conditions of LMCI to generate a simulated dataset. The initial time $t_0$ is 50 and observational interval is one, and the parameters of the system follows: $A_0 = 41.57$, $\tau_{p_0} = 4.21$, $\tau_{o_0} = 28.66$, $N_0 = 0.48$, $C_0 = 6.03$, $\lambda_{A_\beta} = 0.1612$, $K_{A_\beta} = 264.99$, $\lambda_\tau = 0.08$, $K_{\tau_p} = 131.66$, $\lambda_{\tau_o} = 1.74$, $\lambda_{N_{\tau_o}} = 0.000424$, $\lambda_{N_{\tau_p}} = 0.00737$, $K_N = 1.02$, $\lambda_{CN} = 1.26$, $\lambda_{C_\tau} = 1.93$, $K_C = 129.4$. We chose LMCI because it is a risk state that

leads the real Alzheimer's disease. The oracle causal relationship between features is shown in Figure 4(a) (Hao et al., 2022). Compared to the original setting of the Hao model, we add an extra link between $\tau_o$ and $C$ to introduce a confounder structure but do not re-estimate the parameters (Mielke et al., 2017). We discard $\tau_o$ during the experiment process to introduce an unmeasured confounder. Therefore, the dataset has four observable features.

$$
\begin{cases}
\dfrac{dA_\beta}{dt} = \lambda_{A_\beta} A_\beta (1 - \dfrac{A_\beta}{K_{A_\beta}}) \\[2mm]
\dfrac{d\tau_p}{dt} = \lambda_\tau A_\beta (1 - \dfrac{\tau_p}{K_{\tau_p}}) \\[2mm]
\dfrac{d\tau_o}{dt} = \lambda_{\tau_o} \\[2mm]
\dfrac{dN}{dt} = (\lambda_{N_{\tau_o}} \tau_o + \lambda_{N_{\tau_p}} \tau_p)(1 - \dfrac{N}{K_N}) \\[2mm]
\dfrac{dC}{dt} = (\lambda_{CN} N + \lambda_{C\tau}(\tau_p + \tau_o))(1 - \dfrac{C}{K_C})
\end{cases}
\tag{14}
$$

**Zheng Model** (Zheng et al., 2022). This model includes four features (i.e., $A_\beta$, tau protein $\tau$, $N$, and $C$) with the causal structure shown in Figure 4(b). Zheng's model does not contain unmeasured features and all four features are observable. Zheng's model describes the dynamics of LMCI and Alzheimer's disease and its form follows Equation 15. The initial DPS time $t_0$ is -10 and observation interval is two, $A_\beta(t_0) = y_0$, $\tau_p(t_0) = 0$, $\tau_o(t_0) = 0$, $N(t_0) = 0$, and $C(t_0) = 0$ and the parameters follows: $w_{A0} = 0$, $w_{A1} = 0.745$, $w_{A2} = -0.749$, $w_{\tau0} = 0$, $w_{\tau1} = 0.689$, $w_{\tau2} = -0.679$, $w_{\tau3} = 0$, $w_{\tau4} = 0.185$, $w_{\tau5} = -0.101$, $w_{\tau6} = 0$, $w_{N0} = 0$, $w_{N1} = 0.899$, $w_{N2} = -0.927$, $w_{N3} = 0.554$, $w_{N4} = 1.792$, $w_{N5} = -2.127$, $w_{C0} = 0$, $w_{C1} = 0.134$, $w_{C2} = -0.067$, $w_{C3} = 0.004$, $w_{C4} = 0.007$, $w_{C5} = -0.008$, $y_0 = 0.000141$.

$$
\begin{cases}
\dfrac{dA_\beta}{dt} = w_{A1} A_\beta + w_{A2} A_\beta^2 \\[2mm]
\dfrac{d\tau}{dt} = w_{\tau1}\tau + w_{\tau2}\tau^2 + w_{\tau3}A_\beta + w_{\tau4}A_\beta^2 + w_{\tau5}A_\beta\tau \\[2mm]
\dfrac{dN}{dt} = w_{N1}N + w_{N2}N^2 + w_{N3}\tau + w_{N4}\tau^2 + w_{N5}\tau N \\[2mm]
\dfrac{dC}{dt} = w_{C1}C + w_{C2}C^2 + w_{C3}N + w_{C4}N^2 + w_{C5}NC
\end{cases}
\tag{15}
$$

### B.3 MK Dataset

We also generated two Michaelis-Menten kinetics datasets (i.e., MM-25, MM-50) to evaluate the performance of the CTP model in high-dimensional case (Zhang et al., 2022). The original MM-25 contains 25 nodes and the original MM-50 contains 50 nodes. The causal relation and dynamics of the two datasets follow Equation 16, where $X_i$ denotes a node and $N_i$ denotes the set of the parent nodes of $X_i$. The causal relation was from a randomly generated DAG whose edge probability was 0.15, and the vertices followed the topological order. The initial value of each node is a sample of a uniform distribution that takes a value between 0.8 and 1.2. The initial time is zero and the observation interval is 0.25. We discarded the first five nodes to introduce unmeasured confounding. The final MM-25 and MM-50 dataset contains 20 and 45 observable features, respectively.

$$
\frac{dX_i}{dt} = -X_i + \frac{1}{|N_i|} \sum_{j \in N_i} \frac{X_j}{1 + X_j}.
\tag{16}
$$

### B.4 Generation of Simulated datasets

For each model, we generated training, validation, and test datasets with the same process, respectively (Virtanen et al., 2020). Each sample contains 15 observations distributed evenly. We randomly

discarded 5% observations to simulate missing data. We define observations before the eighth observation is used as the input and reconstruction target of the CTP, and observations after the eighth observation are only used for prediction. We normalized all variables to a zero mean unit variance distribution after data data-generating process. Besides the above settings, we generated datasets in three different configurations.

1. All patients share the same init value with parameters of the dynamic system vary. We added uniform noise to the parameters of each patient, the noise follows uniform distribution, and the strength is 10% of the parameters. Therefore, each patient has its disease progression trajectories.

2. All patients share the same parameters of the dynamic system with initial value varying. We added uniform noise to the init values for each patient, the noise follows uniform distribution, and the strength is 10% of the original init value.

3. Each patient has its own parameters and init values. We added uniform noise to the init values and parameters of each patient, the noises follow uniform distribution, and the strength is 10%.

We generated 1,024, 128, and 128 samples with respect to training, validation, and test datasets for the three configurations. We used datasets generated in the third configuration to conduct experiments in the main body. We also generated a training dataset in the third configuration with 128, 256, and 512 samples in the training set. We conducted low resource analysis and robust analysis according to the unused datasets, whose details are in Appendix E.

# C  ALGORITHM

## C.1  CTP MODEL OPTIMIZE

---

**Algorithm 1** CTP Model Optimize Process

---

**Input:** training data $D_t$, validation data $D_v$, max iteration $n$, update interval $m$, convergence threshold $\delta$, Lagrangian parameters $\alpha$, $\rho$, $\eta$, $\gamma$

**Output:** Loss $\mathcal{L}$, parameters $\phi$, $\theta$

    *Initialize* : $\phi$, $\theta$, loss list $a_1$: [], $a_2$: []

  1: **for** $i = 1$ to $n$ **do**
  2:    **if** $i \bmod m = 0$ **then**
  3:       Calculate $\mathcal{L}_{final}$ according Equation 10 given $D_v$
  4:       Calculate $h(\theta)$ according to Equation 7
  5:       Append $\mathcal{L}_{final}$ to loss list $a_1$
  6:       Append $h(\theta)$ to loss list $a_2$
  7:    **end if**
  8:    **if** $(i \bmod m) == 0$ and $i > 2m$ **then**
  9:       **if** $a_1[-1] < a_1[-2]$ **then**
10:          $\Delta\lambda = (a_1[-2] - a_1[-1])/m$
11:          **if** $\Delta\lambda < \delta$ **then**
12:             $h = a_2[-1]$
13:             $\rho = \alpha h + \rho$
14:             **if** $a_2[-1] > a_2[-2] \times \gamma$ **then**
15:                $\alpha = \eta \times \alpha$
16:             **end if**
17:          **end if**
18:       **end if**
19:    **end if**
20:    Sample mini-batch $B$ from $D_t$
21:    Calculate $\mathcal{L}_{final}$ according Equation 10 given $B$
22:    Calculate gradient $\nabla\mathcal{L}_{final}$ w.r.t. $\phi$, $\theta$
23:    Update $\phi$, $\theta$ via a stochastic optimizer
24: **end for**
25: Calculate $\mathcal{L}$ according Equation 8 given $D_v$
26: **return** $\mathcal{L}$, $\phi$, $\theta$

---

Line 2 to 19 describes the details of generating a Lagrangian multiplier, while line 20 to 24 describes a normal gradient-based parameters optimization process. During the optimization process, we calculate and record $\mathcal{L}_{final}$ and $h(\theta)$ every $m$ steps (line 3-7). We regard a subproblem as solved when it converges (lines 9-11). Then, we gradually increase the value of $\alpha$ and $\rho$ (lines 13-15) and establish a new subproblem. Eventually, all connections that violate the DAG assumption between features will be eliminated.

## C.2   CAUSAL IDENTIFICATION

---

**Algorithm 2** Causal Identification

---

**Input:** training data $D_t$, validation data $D_v$, accept ratio $\rho$, model threshold $\delta$, causal threshold $\varphi$,
   model number $N$,
**Output:** Causal Mask $\mathcal{M}$
   *Initialize* : $\mathcal{M}$ $\widetilde{\mathcal{M}}$ via Eq. 11
 1: **while** $\sum_{ij}(\widetilde{\mathcal{M}}_{ij}) > 0$ **do**
 2:    Define model list M_List $= []$
 3:    **while** Length(M_List) $< N$ **do**
 4:       Optimize a new model via Alg. 1, obtain $\mathcal{L}, \phi, \theta$
 5:       Summarize $\widetilde{D}$ according $\theta$
 6:       **if** $\mathcal{L} < \delta$ **then**
 7:          Append $\widetilde{D}$ to M_List
 8:       **end if**
 9:    **end while**
10:    **for** $i = 1, j = 1$ to $K + 1, K + 1$ **do**
11:       **for** $n = 1$ to $N$ **do**
12:          $e_{ij} = 0$
13:          **if** M_List$[n]_{i,j} > \varphi$ **then**
14:             $e_{ij} = e_{ij} + 1$
15:          **end if**
16:       **end for**
17:       **if** $\widetilde{\mathcal{M}}_{ij}$ and $e_{ij}/N > \rho$ **then**
18:          $\mathcal{M}_{ij} = 1$
19:          $\widetilde{\mathcal{M}}_{ij} = 0$
20:       **else if** $\widetilde{\mathcal{M}}_{ij}$ and $e_{ij}/N < (1 - \rho)$ **then**
21:          $\mathcal{M}_{ij} = 0$
22:          $\widetilde{\mathcal{M}}_{ij} = 0$
23:       **end if**
24:    **end for**
25: **end while**
26: **return** $\mathcal{M}$

---

The algorithm first repeatedly optimizes independent CTP models and records $N$ models that can converge successfully (i.e., validation loss $\mathcal{L}$ is less than a predefined threshold $\delta$) (lines 2-8). We presume these models are more probable to identify correct causal relationships. Then, We analyze the neural connectivity matrix of each model. We treat elements whose value is less than a threshold $\varphi$ as invalid (line 13) (Lachapelle et al., 2020). We use the $e_{ij}/N$ to determine how many models treat the connection $i \leftarrow j$ as invalid. If the value is larger than an accept ratio $\rho$, we will treat the connection as certainly valid and set $\mathcal{M}_{ij} = 1$ and $\widetilde{\mathcal{M}}_{ij} = 0$. $e_{ij}/N$ is less than $1 - \rho$ indicates the connection is certainly invalid and we we set $\mathcal{M}_{ij} = 0$ and $\widetilde{\mathcal{M}}_{ij} = 0$ (line 17-23). We repeat the process until all casual relations become certain (i.e., $\sum_{ij}(\widetilde{\mathcal{M}}_{ij}) = 0$))

## C.3 INDEPENDENT CTP MODELS OPTIMIZE

---

**Algorithm 3** independent CTP Models Optimize Process

---

**Input:** $D_t$, $L$, $M^\star$, $\mathrm{lr}_p$, $\mathrm{lr}_t$, $\mathrm{optim}_p$, $\mathrm{optim}_t$
**Output:** A series of new CTP model $\{M_l\}_{l=1}^L$
  *Initialize* : Parameters of $\{M^l\}_{l=1}^L$ via $M^\star$
1: **for** $j = 1$ to $n$ **do**
2:  Sample mini-batch $B$ from $D_t$
3:  Define prediction loss $\mathcal{L}_p = 0$
4:  **for** $l = 1$ to $L$ **do**
5:   Calculate $\mathcal{L}_l$ via Eq. 8 given $B$ and $M_l$
6:   $\mathcal{L}_p = \mathcal{L}_p + \mathcal{L}_l$
7:  **end for**
8:  Define treatment loss $\mathcal{L}_{treat} = 0$
9:  Define estimation list $\mathrm{EL} = []$
10:  Randomly sample $t_a > t_0$ and $t_o > t_a$
11:  **for** $l = 1$ to $L$ **do**
12:   Apply treatment $\mathrm{do}(A^{t_a} = a)$ to $M_l$
13:   **for** $i = 1$ to $|B|$ **do**
14:    Calculate ${}^l\hat{v}_i'$ via Eq. 5 given $M^l$ and $s_i$ from $B$
15:    Append ${}^l\hat{v}_i(t_o)$ to $\mathrm{EL}$
16:   **end for**
17:  **end for**
18:  Compute pairwise distance $\mathcal{L}_t$ of $\mathrm{EL}$ via Eq. 12
19:  Calculate gradient $\nabla \mathcal{L}_p$, and $\nabla \mathcal{L}_t$
20:  Minimize $\mathcal{L}_p$ via $\nabla \mathcal{L}_p$, $\mathrm{optim}_p$, and $\mathrm{lr}_p$
21:  Maximize $\mathcal{L}_t$ via $\nabla \mathcal{L}_t$, $\mathrm{optim}_t$, and $\mathrm{lr}_t$
22: **end for**
23: **return** $\{M^l\}_{l=1}^L$

---

We use $\mathrm{do}(A^{t_a} = a)$ to denote a treatment, which means fixing the value of feature $\bar{v}^A$ to $a$ from a time point $t_a$ (line 12), regardless its original value. Given a patient data $s_i$ (from $B$) and a model $M^l$, we can implement treatment effect prediction by performing $\mathrm{do}(A^{t_a} = a)$. Then, we use an ODE solver to solve the ODEs, and the solver will generate progression trajectories ${}^l\hat{v}_i(t)$ under a treatment (line 14). Of note, $M^l$ evaluates treatment effect appropriately. By utilizing the causal mask $\mathcal{M}$ from $M^\star$, we can access the causal relationship between features and ensure the dynamics of ancestral features of $A$ will not be affected. Meanwhile, $M^l$ infers the value of the unmeasured confounder so that we can adjust the confounding bias. We record the patient characteristics ${}^l\hat{v}_i(t_o)$ at a randomly selected time point $t_o$ (line 15). To make trajectories of $M^l$ as dissimilar as possible, we maximize the pair-wise distance between recorded ${}^l\hat{v}_i(t_o)$ (Equation 12). In this study, we applied the simplest p-norm distance for computational efficiency. Then, we alternatively maximize the $\mathcal{L}_t$ and minimize the $\mathcal{L}_p$ to keep the group of models can predict trajectories accurately but behave as differently as possible under a treatment.

# D  DISCUSSION

We investigate the performance of the CTP and baselines in trajectory prediction, causal discovery, and treatment effect prediction. Experimental results demonstrate that our CTP model obtains SOTA performance or comparable performance to SOTA in trajectory prediction task, indicating the incorporation of additional penalties does not decrease its prediction performance. The CTP model obtains significantly better causal discovery performance by incorporating ridge loss, while the original neural ODE, as well as TE-CDE and CF-ODE, cannot learn the causal relationship well between features. Meanwhile, its causal discovery performance is further improved by utilizing the DAG penalty. The advantages in extracting causal relationships help the CTP model obtain significantly better performance compared to all baselines in the successive treatment effect prediction test. We also find that the bound generated by the CTP model can include the true trajectories when unmeasured confounder exists, indicating the model may help clinical decisions for complex NCDs.

The main strength of the CTP is that it can predict disease progression trajectories under a treatment. Prognostic analysis aims to assist clinical decision-making and improve prognosis (Gill, 2012). This goal is usually mistakenly understood as developing accurate prediction models. As different NCD patients usually require different treatments due to patient heterogeneity, merely predicting poor prognosis in an unexplainable manner is not enough to assist clinicians in implementing treatment in advance. Clinicians criticized that recent ML-based prognostic models did not help them to deliver more precise treatment, even these models obtained impressive prediction performance Wilkinson et al. (2020). Although researchers have tried to propose explainable models, these models have their shortages. Existing explainable models still rely on capturing correlational relations within features and merely generating local explanations Loh et al. (2022). They are unlikely can be applied in answering counterfactual questions Kim et al. (2020). Our experiment result also demonstrats that the performance of baselines falls sharply under a treatment, indicating they cannot be utilized in treatment effect analysis. We argue that a model can assist clinical decision-making when it can answer counterfactual questions. Compared to existing treatment effect models which only investigate the treatment of a predefined drug to a predefined static outcome, our CTP model can predict the dynamics of the entire system under a treatment Yao et al. (2021). Of note, the progression of NCD is usually a longitudinal, complex process that cannot be comprehensively described by the occurrence of a static binary outcome. Therefore, our CTP model may be deployed in clinical workflow to assist clinicians in delivering better prescriptions.

This study has two advantages and two shortcomings. The first advantage is that we have fully evaluated the model's performance in causal discovery and trajectory prediction. We also quantitatively and qualitatively evaluated the model's performance in treatment effect analysis. It comprehensively explains that our model can be used for prognostic prediction and treatment effect analysis of long-term disease progression trajectories. Previous studies usually do not focus on predicting an entire trajectory, but only on the prognosis at a single isolated time point. Previous research generally deals with causal discovery and treatment effect analysis separately, with few studies completing a full inference of causal discovery and treatment effect analysis. The second advantage is that we conducted comprehensive experiments. In this study, we chose a real public clinical dataset, two simulated Alzheimer's disease amyloid deposition datasets, and two completely simulated high-dimensional datasets for analysis. At the same time, we chose five related models published in recent years as baselines, fully proving the superior performance and universality of the CTP model. This study also has two shortcomings. The first shortcoming is that we assumed that the causal structure of the data constitutes a DAG, which means that our model cannot handle feedback phenomena in dynamic time series systems. The second shortcoming is that we only conducted complete experiments on simulated datasets. In future research, we will try to evaluate the performance of model results in causal analysis on real datasets.

# E EXTENDED EXPERIMENT RESULT

## E.1 TRAJECTORY PREDICTION PERFORMANCE ON SIMULATED DATASETS

Table 6 describes the performances of the CTP and baselines in four simulated datasets. Our CTP model obtained comparable performance to SOTA models.

| Model | Dataset | Reconstruct | Predict | Dataset | Reconstruct | Predict |
|---|---|---|---|---|---|---|
| LODE | Hao | 0.10±0.01 | 0.12±0.03 | MM-25 | 0.04±0.00 | 0.26±0.02 |
| NODE | Hao | **0.01±0.00** | **0.01±0.00** | MM-25 | **0.02±0.00** | 0.25±0.01 |
| NGM | Hao | 0.02±0.01 | 0.02±0.01 | MM-25 | 0.03±0.01 | 0.27±0.01 |
| TE-CDE | Hao | 0.02±0.01 | **0.01±0.00** | MM-25 | **0.02±0.00** | 0.25±0.01 |
| CF-ODE | Hao | 0.02±0.00 | **0.01±0.00** | MM-25 | **0.02±0.00** | **0.23±0.01** |
| CTP | Hao | **0.01±0.01** | 0.02±0.00 | MM-25 | 0.04±0.01 | 0.24±0.01 |
| LODE | Zheng | 0.12±0.03 | 0.15±0.02 | MM-50 | 0.03±0.00 | **0.24±0.00** |
| NODE | Zheng | **0.03±0.01** | **0.08±0.02** | MM-50 | **0.02±0.00** | 0.26±0.01 |
| NGM | Zheng | 0.09±0.01 | 0.11±0.02 | MM-50 | **0.02±0.01** | 0.25±0.01 |
| TE-CDE | Zheng | 0.09±0.01 | 0.11±0.01 | MM-50 | 0.03±0.00 | 0.27±0.02 |
| CF-ODE | Zheng | 0.09±0.02 | 0.11±0.01 | MM-50 | 0.03±0.01 | 0.26±0.01 |
| CTP | Zheng | 0.09±0.01 | 0.09±0.01 | MM-50 | 0.04±0.01 | 0.28±0.02 |

Table 6: Simulated Data Prediction Performance

## E.2 INCLUSION RATE

We evaluated whether a true feature value is included in the estimated bound using the inclusion rate (Figure 5). We found that the inclusion rate is relevant to the number of CTP models. The more the number of models, the better the performance. For example, the inclusion rate of the Hao dataset is 0.22 when we only use two models, while it gradually increases to 0.71 when we apply 64 models. The inclusion rate on other datasets also shows a similar tendency. Therefore, we argue that our method is able to estimate the possible bounds of a treatment, which will assist in better clinical decisions. It is worth noting that different dataset requires different number of independent models. The inclusion rate converges when the model number is larger than 32 in the Hao dataset, while it converges when the model number is larger than eight in the Zheng dataset. The inclusion rate of the MM-25 dataset and MM-50 dataset is significantly lower than the other two datasets.

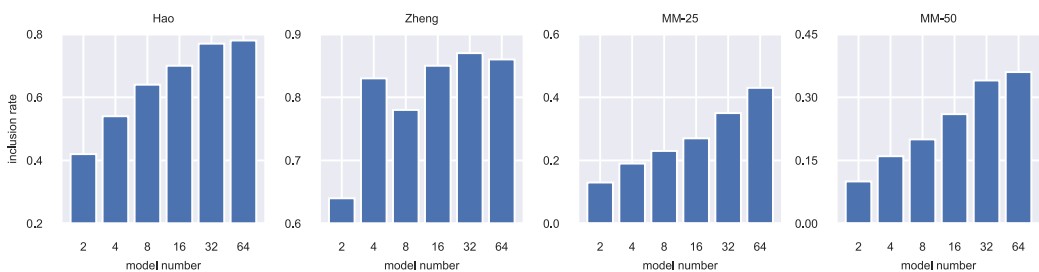

Figure 5: Treatment Inclusion

## E.3 LOW RESOURCE ANALYSIS

Table 7 describes the performance of the CTP and baselines with different sizes of training data. Our CTP model obtained similar performance compared to baselines, though its performance is not the best. We did not observe significant performance deterioration, which indicates all models can successfully converge by feeding limited training data.

| Model | Sample | Dataset | Reconstruct | Predict | Dataset | Reconstruct | Predict |
|---|---|---|---|---|---|---|---|
| LODE | 128 | Hao | 0.11±0.03 | 0.13±0.04 | MM-25 | 0.05±0.01 | 0.21±0.01 |
| NODE | 128 | Hao | **0.01**±**0.00** | **0.02**±**0.01** | MM-25 | **0.04**±**0.00** | 0.25±0.00 |
| NGM | 128 | Hao | 0.02±0.00 | 0.03±0.01 | MM-25 | **0.04**±**0.00** | **0.23**±**0.00** |
| TE-CDE | 128 | Hao | 0.03±0.01 | 0.02±0.00 | MM-25 | **0.04**±**0.01** | 0.24±0.00 |
| CF-ODE | 128 | Hao | 0.02±0.00 | 0.03±0.01 | MM-25 | 0.06±0.00 | **0.23**±**0.01** |
| CTP | 128 | Hao | 0.02±0.00 | 0.03±0.01 | MM-25 | **0.04**±**0.01** | 0.24±0.00 |
| LODE | 128 | Zheng | 0.15±0.03 | 0.23±0.04 | MM-50 | 0.04±0.00 | 0.24±0.01 |
| NODE | 128 | Zheng | **0.04**±**0.01** | **0.05**±**0.01** | MM-50 | **0.03**±**0.01** | 0.25±0.01 |
| NGM | 128 | Zheng | 0.07±0.02 | 0.08±0.02 | MM-50 | **0.03**±**0.01** | 0.24±0.00 |
| TE-CDE | 128 | Zheng | 0.10±0.01 | 0.11±0.01 | MM-50 | 0.04±0.00 | **0.23**±**0.01** |
| CF-ODE | 128 | Zheng | 0.08±0.02 | 0.12±0.03 | MM-50 | 0.04±0.00 | 0.26±0.00 |
| CTP | 128 | Zheng | 0.09±0.01 | 0.11±0.02 | MM-50 | 0.08±0.00 | 0.30±0.00 |
| LODE | 256 | Hao | 0.09±0.01 | 0.08±0.02 | MM-25 | 0.05±0.00 | **0.22**±**0.01** |
| NODE | 256 | Hao | **0.01**±**0.00** | 0.03±0.01 | MM-25 | **0.03**±**0.01** | 0.23±0.00 |
| NGM | 256 | Hao | 0.02±0.01 | 0.04±0.00 | MM-25 | **0.03**±**0.00** | 0.23±0.00 |
| TE-CDE | 256 | Hao | **0.01**±**0.00** | **0.02**±**0.00** | MM-25 | 0.05±0.00 | 0.24±0.00 |
| CF-ODE | 256 | Hao | **0.01**±**0.00** | 0.03±0.01 | MM-25 | 0.04±0.00 | **0.22**±**0.02** |
| CTP | 256 | Hao | 0.02±0.00 | 0.03±0.01 | MM-25 | 0.04±0.01 | 0.23±0.00 |
| LODE | 256 | Zheng | 0.12±0.03 | 0.17±0.03 | MM-50 | **0.04**±**0.00** | **0.24**±**0.00** |
| NODE | 256 | Zheng | **0.05**±**0.01** | **0.06**±**0.01** | MM-50 | 0.02±0.00 | **0.24**±**0.00** |
| NGM | 256 | Zheng | 0.07±0.01 | 0.08±0.01 | MM-50 | **0.04**±**0.01** | 0.24±0.01 |
| TE-CDE | 256 | Zheng | 0.08±0.02 | 0.11±0.02 | MM-50 | 0.06±0.01 | 0.25±0.00 |
| CF-ODE | 256 | Zheng | 0.09±0.02 | 0.10±0.01 | MM-50 | 0.05±0.00 | **0.24**±**0.00** |
| CTP | 256 | Zheng | 0.09±0.03 | 0.10±0.01 | MM-50 | 0.07±0.00 | 0.26±0.00 |
| LODE | 512 | Hao | 0.10±0.01 | 0.09±0.01 | MM-25 | 0.05±0.00 | 0.23±0.00 |
| NODE | 512 | Hao | **0.01**±**0.00** | **0.02**±**0.01** | MM-25 | **0.03**±**0.00** | 0.22±0.00 |
| NGM | 512 | Hao | 0.02±0.00 | 0.04±0.01 | MM-25 | 0.04±0.00 | 0.22±0.02 |
| TE-CDE | 512 | Hao | 0.02±0.00 | 0.03±0.00 | MM-25 | **0.03**±**0.00** | 0.23±0.00 |
| CF-ODE | 512 | Hao | 0.03±0.01 | 0.03±0.01 | MM-25 | 0.04±0.00 | 0.23±0.01 |
| CTP | 512 | Hao | 0.02±0.00 | 0.03±0.01 | MM-25 | **0.03**±**0.00** | **0.21**±**0.00** |
| LODE | 512 | Zheng | 0.10±0.01 | 0.16±0.02 | MM-50 | 0.04±0.01 | **0.24**±**0.01** |
| NODE | 512 | Zheng | **0.05**±**0.01** | **0.05**±**0.01** | MM-50 | **0.03**±**0.01** | 0.25±0.00 |
| NGM | 512 | Zheng | 0.08±0.02 | 0.10±0.02 | MM-50 | **0.03**±**0.01** | 0.26±0.00 |
| TE-CDE | 512 | Zheng | 0.08±0.01 | 0.11±0.02 | MM-50 | **0.03**±**0.00** | 0.26±0.01 |
| CF-ODE | 512 | Zheng | 0.09±0.02 | 0.10±0.01 | MM-50 | **0.03**±**0.00** | 0.28±0.00 |
| CTP | 512 | Zheng | 0.08±0.01 | 0.11±0.01 | MM-50 | 0.06±0.00 | 0.29±0.01 |

Table 7: Low Resource Prediction Performance

Table 8 describes the causal discovery performance of the CTP and baselines with different sizes of training data, where we only recorded AUC for clarity. We also observe that the size of the training data did not influence the causal discovery performance significantly. The NGM and LODE obtained significantly better performance because they utilized ridge loss to remove unnecessary relations between features. Our CTP model obtained better performance compared to the NGM, which may be attributed to the incorporation of score-based DAG constraints.

| Model | Sample | Hao | Zheng | MM-25 | MM-50 |
|---|---|---|---|---|---|
| LODE | 128 | 0.64±0.01 | 0.60±0.00 | 0.82±0.02 | **0.90±0.00** |
| NODE | 128 | 0.55±0.00 | 0.52±0.00 | 0.57±0.00 | 0.55±0.01 |
| NGM | 128 | 0.69±0.01 | 0.56±0.00 | 0.67±0.01 | 0.63±0.00 |
| TE-CDE | 128 | 0.54±0.01 | 0.52±0.01 | 0.55±0.00 | 0.53±0.01 |
| CF-ODE | 128 | 0.53±0.01 | 0.53±0.01 | 0.53±0.01 | 0.56±0.00 |
| CTP | 128 | **0.78±0.02** | **0.66±0.00** | **0.86±0.01** | **0.90±0.01** |
| LODE | 256 | 0.68±0.01 | 0.56±0.00 | 0.83±0.00 | **0.90±0.00** |
| NODE | 256 | 0.57±0.02 | 0.51±0.01 | 0.54±0.01 | 0.54±0.00 |
| NGM | 256 | 0.70±0.01 | 0.61±0.00 | 0.62±0.01 | 0.67±0.01 |
| TE-CDE | 256 | 0.56±0.01 | 0.53±0.01 | 0.54±0.01 | 0.54±0.01 |
| CF-ODE | 256 | 0.54±0.02 | 0.54±0.00 | 0.53±0.01 | 0.53±0.01 |
| CTP | 256 | **0.79±0.01** | **0.64±0.01** | **0.87±0.02** | **0.90±0.01** |
| LODE | 512 | 0.65±0.02 | 0.53±0.02 | 0.82±0.04 | 0.89±0.01 |
| NODE | 512 | 0.55±0.01 | 0.52±0.00 | 0.58±0.01 | 0.53±0.02 |
| NGM | 512 | 0.67±0.00 | 0.60±0.01 | 0.62±0.01 | 0.63±0.01 |
| TE-CDE | 512 | 0.58±0.00 | 0.51±0.00 | 0.53±0.01 | 0.52±0.01 |
| CF-ODE | 512 | 0.54±0.01 | 0.53±0.01 | 0.55±0.00 | 0.57±0.01 |
| CTP | 512 | **0.81±0.00** | **0.63±0.01** | **0.86±0.00** | **0.91±0.01** |

Table 8: Low Resource Causal Discovery Performance

Table 9 describes the treatment effect prediction performance of the CTP and baselines with different sizes of training data, where we only recorded full performance for clarity. Our CTP model obtained better performance compared to baselines, and we also found that the data size does not influence model performance significantly.

| Model | Sample | Hao | Zheng | MM-25 | MM-50 |
|---|---|---|---|---|---|
| LODE | 128 | 0.84±0.15 | 0.87±0.07 | 1.33±0.04 | 1.49±0.03 |
| NODE | 128 | 1.45±0.23 | 2.16±0.02 | 1.28±0.04 | 1.52±0.02 |
| NGM | 128 | 0.26±0.02 | 0.83±0.02 | 0.86±0.02 | 2.40±0.01 |
| TE-CDE | 128 | 0.40±0.02 | 3.65±0.02 | 1.24±0.07 | 1.52±0.02 |
| CF-ODE | 128 | 0.54±0.01 | 0.38±0.03 | 1.39±0.07 | 1.58±0.06 |
| CTP | 128 | **0.24±0.02** | **0.34±0.01** | **0.83±0.04** | **1.19±0.06** |
| LODE | 256 | 0.78±0.13 | 0.84±0.05 | 1.27±0.04 | 1.53±0.04 |
| NODE | 256 | 1.40±0.17 | 2.37±0.02 | 1.20±0.06 | 1.50±0.02 |
| NGM | 256 | 0.24±0.02 | 0.75±0.04 | 0.88±0.04 | 2.41±0.01 |
| TE-CDE | 256 | 0.40±0.02 | 3.53±0.02 | 1.32±0.04 | 1.49±0.02 |
| CF-ODE | 256 | 0.65±0.03 | 0.42±0.03 | 1.28±0.05 | 1.55±0.05 |
| CTP | 256 | **0.22±0.01** | **0.39±0.04** | **0.80±0.03** | **1.17±0.05** |
| LODE | 512 | 0.81±0.08 | 0.80±0.05 | 1.21±0.05 | 1.52±0.03 |
| NODE | 512 | 1.37±0.12 | 2.09±0.01 | 1.23±0.03 | 1.51±0.01 |
| NGM | 512 | 0.23±0.01 | 0.80±0.01 | 0.89±0.03 | 2.42±0.02 |
| TE-CDE | 512 | 0.36±0.02 | 3.67±0.03 | 1.25±0.04 | 1.51±0.02 |
| CF-ODE | 512 | 0.57±0.02 | 0.40±0.03 | 1.28±0.05 | 1.54±0.04 |
| CTP | 512 | **0.19±0.01** | **0.35±0.04** | **0.78±0.04** | **1.16±0.08** |

Table 9: Low Resource Treatment Effect Prediction

## E.4 ROBUSTNESS ANALYSIS

Table 10 describes the performance of the CTP and baselines in different configurations. The NODE model obtains the best performance, which is not surprising and consists of other experiment results. Our CTP models, as well as other baselines, obtained slightly worse performance. However, the performance of our CTP model and other baselines are also acceptable.

| Model | Setting | Dataset | Predict | Reconstruct | Dataset | Predict | Reconstruct |
|---|---|---|---|---|---|---|---|
| LODE | 1 | Hao | 0.06±0.01 | 0.04±0.00 | MM-25 | 0.06±0.01 | 0.22±0.00 |
| NODE | 1 | Hao | **0.02±0.00** | **0.03±0.01** | MM-25 | **0.02±0.00** | **0.21±0.01** |
| NGM | 1 | Hao | 0.04±0.01 | 0.04±0.00 | MM-25 | 0.03±0.01 | **0.21±0.00** |
| TE-CDE | 1 | Hao | 0.04±0.01 | 0.04±0.01 | MM-25 | 0.04±0.01 | 0.23±0.02 |
| CF-ODE | 1 | Hao | 0.04±0.01 | **0.03±0.00** | MM-25 | 0.05±0.01 | 0.22±0.01 |
| CTP | 1 | Hao | 0.03±0.00 | **0.03±0.01** | MM-25 | 0.06±0.01 | 0.34±0.02 |
| LODE | 1 | Zheng | 0.15±0.03 | 0.14±0.03 | MM-50 | 0.05±0.01 | 0.28±0.02 |
| NODE | 1 | Zheng | **0.07±0.01** | **0.07±0.01** | MM-50 | **0.02±0.00** | 0.23±0.00 |
| NGM | 1 | Zheng | 0.11±0.02 | 0.10±0.02 | MM-50 | 0.03±0.00 | 0.24±0.01 |
| TE-CDE | 1 | Zheng | 0.10±0.01 | 0.11±0.03 | MM-50 | 0.03±0.00 | 0.24±0.00 |
| CF-ODE | 1 | Zheng | 0.12±0.03 | 0.11±0.02 | MM-50 | 0.05±0.00 | 0.26±0.00 |
| CTP | 1 | Zheng | 0.11±0.01 | 0.11±0.02 | MM-50 | 0.07±0.01 | **0.22±0.01** |
| LODE | 2 | Hao | 0.06±0.01 | 0.08±0.01 | MM-25 | 0.05±0.01 | 0.23±0.01 |
| NODE | 2 | Hao | **0.01±0.00** | **0.03±0.01** | MM-25 | **0.02±0.00** | 0.22±0.01 |
| NGM | 2 | Hao | 0.03±0.01 | 0.04±0.00 | MM-25 | 0.04±0.00 | 0.22±0.01 |
| TE-CDE | 2 | Hao | 0.03±0.00 | **0.03±0.01** | MM-25 | 0.04±0.01 | **0.21±0.02** |
| CF-ODE | 2 | Hao | 0.04±0.02 | **0.03±0.01** | MM-25 | 0.04±0.00 | 0.23±0.01 |
| CTP | 2 | Hao | 0.03±0.01 | **0.03±0.01** | MM-25 | 0.07±0.02 | 0.25±0.01 |
| LODE | 2 | Zheng | 0.23±0.06 | 0.18±0.02 | MM-50 | 0.04±0.01 | 0.27±0.02 |
| NODE | 2 | Zheng | **0.08±0.02** | **0.06±0.00** | MM-50 | **0.03±0.01** | **0.24±0.01** |
| NGM | 2 | Zheng | 0.09±0.03 | 0.09±0.01 | MM-50 | **0.03±0.00** | **0.24±0.00** |
| TE-CDE | 2 | Zheng | 0.10±0.01 | 0.10±0.01 | MM-50 | 0.04±0.01 | **0.24±0.02** |
| CF-ODE | 2 | Zheng | 0.09±0.01 | 0.09±0.02 | MM-50 | **0.03±0.00** | 0.25±0.01 |
| CTP | 2 | Zheng | 0.11±0.02 | 0.11±0.03 | MM-50 | 0.08±0.01 | 0.28±0.01 |

Table 10: Robustness Prediction Performance

Table 11 describes the causal discovery performance of the CTP and baselines in different configurations. Our CTP model obtained better performance compared to all other baselines, and we did not find significant performance degradation in the two data generation configurations. The result indicates our model is robust and can discover causal relations from noisy data.

| Model | Setting | Hao | Zheng | MM-25 | MM-50 |
|---|---|---|---|---|---|
| LODE | 1 | 0.67±0.01 | 0.61±0.02 | 0.82±0.02 | **0.90±0.01** |
| NODE | 1 | 0.57±0.01 | 0.51±0.01 | 0.51±0.00 | 0.50±0.01 |
| NGM | 1 | 0.71±0.01 | 0.60±0.01 | 0.63±0.01 | 0.64±0.00 |
| TE-CDE | 1 | 0.55±0.00 | 0.57±0.01 | 0.64±0.01 | 0.58±0.01 |
| CF-ODE | 1 | 0.62±0.00 | 0.55±0.01 | 0.56±0.02 | 0.60±0.02 |
| CTP | 1 | **0.81±0.01** | **0.63±0.01** | **0.86±0.01** | **0.90±0.01** |
| LODE | 2 | 0.69±0.01 | 0.60±0.01 | 0.83±0.02 | 0.90±0.01 |
| NODE | 2 | 0.62±0.00 | 0.51±0.01 | 0.50±0.01 | 0.51±0.01 |
| NGM | 2 | 0.67±0.00 | 0.60±0.01 | 0.62±0.02 | 0.63±0.01 |
| TE-CDE | 2 | 0.58±0.01 | 0.56±0.01 | 0.57±0.03 | 0.53±0.00 |
| CF-ODE | 2 | 0.63±0.01 | 0.55±0.01 | 0.60±0.01 | 0.55±0.01 |
| CTP | 2 | **0.82±0.00** | **0.64±0.01** | **0.87±0.01** | **0.90±0.00** |

Table 11: Robustness Causal Discovery Performance

Table 12 describes the treatment effect prediction performance of the CTP and baselines in different configurations. Our CTP model obtained better performance compared to all other baselines, and we did not find significant performance degradation in the two data generation configurations.

| Model | Setting | Hao | Zheng | MM-25 | MM-50 |
|---|---|---|---|---|---|
| LODE | 1 | 0.82±0.10 | 0.84±0.04 | 1.26±0.03 | 1.57±0.01 |
| NODE | 1 | 1.33±0.17 | 1.98±0.02 | 1.25±0.04 | 1.43±0.02 |
| NGM | 1 | 0.22±0.02 | 0.73±0.03 | 0.83±0.03 | 2.09±0.03 |
| TE-CDE | 1 | 0.35±0.03 | 3.21±0.03 | 1.34±0.05 | 1.65±0.02 |
| CF-ODE | 1 | 0.58±0.02 | 0.38±0.03 | 1.04±0.04 | 1.66±0.04 |
| CTP | 1 | **0.18±0.01** | **0.23±0.04** | **0.81±0.02** | **1.11±0.04** |
| LODE | 2 | 0.97±0.21 | 0.77±0.03 | 1.30±0.02 | 1.49±0.02 |
| NODE | 2 | 1.04±0.09 | 1.79±0.03 | 1.21±0.05 | 1.47±0.01 |
| NGM | 2 | 0.24±0.01 | 0.69±0.04 | 0.87±0.02 | 1.99±0.04 |
| TE-CDE | 2 | 0.36±0.03 | 3.03±0.04 | 1.23±0.04 | 1.59±0.03 |
| CF-ODE | 2 | 0.59±0.04 | 0.35±0.02 | 0.94±0.01 | 1.69±0.07 |
| CTP | 2 | **0.15±0.01** | **0.24±0.02** | **0.87±0.06** | **1.02±0.06** |

Table 12: Robustness Treatment Effect Prediction

