# OpenReview forum: "CTP: A Causal Interpretable Model for Non-Communicable Disease Progression Prediction"
_ICLR.cc/2024/Conference — ICLR 2024 Conference Withdrawn Submission_

### Official Review · Reviewer_Bw5q · 2023-10-30

**Soundness:** 2 fair
**Presentation:** 1 poor
**Contribution:** 3 good
**Rating:** 5
**Confidence:** 3

**Summary:**

The objectives of this work are to recover true causal relationships within medical time series and thus to improve model interpretability.

The method proposed relies on Neural ODEs to model relationships between variables. The product of NN weight matrices is taken as their causal adjacency matrix. Two regularizations are introduced to improve the quality of the extracted causal relationships: L1 regularization and a ‘DAG loss’ that ensures the resulting causal graph has no cyclic dependencies.

Authors propose an additional iterative retraining procedure to progressively improve confidence in learned causal relationships, by measuring agreement within an ensemble of models.

Finally, an ensemble of CTP models is used to measure uncertainty with respect to possible treatment effects (e.g. due to unobserved confounding). These are retrained to maximize their difference in predicting treatment effects.

The model is studied on different simulated and real datasets.

**Strengths:**

*  Recovering causal relationships in time-series models is an important and challenging problem.
* An sensible method is proposed, with relevant ideas: enforcing sparsity, avoiding cyclic dependencies, measuring treatment effect bounds.
* Promising experimental results on simulations: causal identification does not harm test prediction performance and can identify causal relationships, which leads to better estimates of counterfactual trajectories.

**Weaknesses:**

* The paper reads as a collection of complex modeling steps which do not appear always well-motivated or ablated in experimental results.
  * With the main focus of the paper being on causal discovery, there are little to no guarantees. See Q about graph directionality below.
  * Are the small gains achieved by CTP on top of Linear ODE + L1 due to non-linearity or to the DAG loss term? NGM does not help answer this as it does not have the same L1 regularization.
  * No numerical experiments measuring the quality of treatment effect bounds. I am surprised they are "not very loose", since they are maximized in training.
  * Finally, I think a figure summarizing the many modeling elements would help the reader.

* No discussion on interpretability, despite it being stated as one of the main motivations behind the paper.
* No discussion of related works. I don't think this should be relegated to the appendix without even a mention in the main text.

* The paper is often confusing and challenging to read. Issues include language, heavy/undefined notation. Some non-exhaustive examples:
  * Paragraph after Eq.7. I would change the $k$ superscript there to another letter to avoid confusion with feature indices used above.
  * please carefully proofread language (“Once the constrained hold”, “proofed”, “adjacent matrix”, "casual", etc.).
  * First paragraph of the introduction could be improved and better connected to the actual content of the paper.
  * Eq 11 has a tuple on the LHS and a single value on the RHS, as far as I understand. Could authors please clarify?
  * See q's about Sec 2.4. below.

**Questions:**

* Sec 2.2:
  * How does $(1-I) \odot \tilde{D}$ become the adjacency matrix of the DAG of interest, i.e. what does the $(1-I)$ achieve?
  * "$A_{ij}^k$ indicates a weighted path count": weighted by what?

* Sec 2.4:
  * "we presume all these parameters are located in a connected region.": what does this mean, and where is this used in your analysis?
  * "To make trajectories of $M^l$ as dissimilar as possible": do authors not mean making trajectories generated by *different* models as dissimilar as possible?

* If the directionality of causal relationships is not recoverable, the ensembling approach proposed should not help recover the true causal graph (e.g. 50% probability of recovering A->B and B->A, for variables A and B). Could authors comment on this potential failure mode and what empirical bounds are then obtained in treatment effect prediction?

* How expensive is the procedure of training multiple CTP models until convergence for causal identification, then again multiple converged models for TE bound estimation? Would this realistically be feasible in practice?

* Is CTP robust to high-dimensional sources of confounding?

* How is missing data (incl. the hidden confounder dimension) treated within the LSTM network?

* What is the pattern for putting results in bold (not always best result)? Overlapping results should be treated in the same way. Same in your analysis: CTP does not obtain “first place” if results overlap.

* Eq 8 (loss): would it ever make sense to weigh the contributions of different features differently? e.g. if they are measured at different frequencies?

---

### Official Review · Reviewer_Z9w3 · 2023-10-31

**Soundness:** 2 fair
**Presentation:** 2 fair
**Contribution:** 2 fair
**Rating:** 5
**Confidence:** 3

**Summary:**

The manuscript discusses a model called Causal Trajectory Prediction (CTP) that aims to predict the disease progression trajectory of patients by controlling a feature based on observational data. The model attempts to address the issue of unmeasured confounders and provides a causal interpretable approach. The study consists of two phases: first, it predicts feature progression trajectories using neural networks, and second, it estimates the bounds of treatment effects. The CTP model is evaluated using both real medical data and simulated datasets. The results indicate that the CTP model performs well in trajectory prediction, causal discovery, and treatment effect prediction.

**Strengths:**

The CTP model offers a novel approach to predicting disease progression trajectories while considering causality and the effects of treatment. It combines trajectory prediction and causal graph discovery, making it unique and potentially valuable for clinical decision-making. The model's ability to handle unmeasured confounders is a significant strength, I believe it is an important consideration in medical data analysis and can lead to more reliable predictions. The experimental results suggest that the CTP model performs well in trajectory prediction, causal discovery, and treatment effect prediction, outperforming or at least matching existing models and, the model's scalability is demonstrated by its good performance in high-dimensional scenarios, such as the MM-25 and MM-50 datasets.

**Weaknesses:**

The model's complexity may be a limitation for some users, particularly those without a strong background in machine learning and causal inference. While the model's performance is evaluated using simulated datasets and real medical data, the absence of real clinical data for treatment effect prediction limits its practical applicability in a clinical setting. Also, the causal discovery performance is only evaluated on simulated datasets, and it would be beneficial to include real clinical data for a more comprehensive assessment.

**Questions:**

Authors should consider elaborating more on the practicality, applicability, and potential limitations of the CTP model to encourage the significance of application in the healthcare domain.
1) The manuscript provides comparisons with existing models, but what are the key differentiators that make the CTP model a superior choice for clinical applications compared to these existing models?
2) While the manuscript demonstrates scalability, are there any limitations or challenges when applying the CTP model to even larger and more complex datasets, which are common in clinical research?
3) How does the model handle situations where the causal assumptions do not hold, or where there may be feedback loops in the underlying biological or clinical processes?
4) What are the computational and resource requirements for implementing this model in a clinical environment, and is it feasible for real-time decision support?
5) How well does the model perform when applied to diverse and real-world clinical datasets with variations in data quality, patient demographics, and healthcare settings? The manuscript mainly focuses on simulated datasets and one real dataset, but real clinical data can be highly heterogeneous.

---

### Official Review · Reviewer_8VqP · 2023-10-31

**Soundness:** 2 fair
**Presentation:** 2 fair
**Contribution:** 2 fair
**Rating:** 3
**Confidence:** 5

**Summary:**

The paper introduces a causal interpretable model designed to predict a patient’s disease progression trajectory. This model is constructed using neural ordinary differential equations and ensures causality through the incorporation of both a sparse penalty and a score-based penalty. The authors have conducted experiments using both synthetic and real-world data, demonstrating the model's performance in causal discovery, trajectory prediction, and treatment effect prediction.

**Strengths:**

- The authors have made their code available, enhancing the reproducibility of their work.

**Weaknesses:**

- The paper employs a static causal graph (Figure 1), despite the data being longitudinal and observational. This raises concerns about the model's capability to model time-varying causal relationships. Did the authors assume that causal relationships remain constant over time? Otherwise, one might consider more advanced models that can capture changes in causality over time.

- There is no clear theoretical analysis provided to ensure the correct identification of the causal graph with the proposed method from observational data.

- The manuscript assumes the existence of unobserved confounding variables but does not provide further sensitivity analysis to quantitatively assess the impact of these variables on estimation uncertainty.

- The evaluation of real-world data is limited, with only one dataset (ADNI) used, and a small sample size of 275. This raises questions about the model's generalizability and effectiveness in practical scenarios.

- The predictive performance on the real-world dataset, as shown in Table 2, is suboptimal (AUC ~ 0.55, close to random guess). Given such results, how can the model perform well on the other two evaluation tasks, which are even more difficult than trajectory prediction?

- The treatment settings are vague:
    - The criteria for treatment selection in each dataset are not provided.
    - The paper’s claim that "a treatment can be direct, dynamic, and continuous" is not substantiated with examples from the datasets (i.e., only one variable at only one-time point as the treatment).
    - The outcomes of interest and the confounding variables in each dataset are not specified.

- While the paper claims that the model can aid in clinical decision-making, a concrete example or case study illustrating this application is needed.

- The presentation of results could be improved for clarity. For instance, in Table 4, only the top-performing results should be highlighted in bold. Figure 2 requires clarification on what is meant by "normalized value" on the y-axis. Missing standard deviations of CTP* in Table 2.

**Questions:**

- The above section.
- How is the time-varying causality in longitudinal data handled?
- How is the identification of the causal graph theoretically guaranteed?
- How does the model assist in clinical decision-making in practice?

---

### Official Review · Reviewer_kBz9 · 2023-11-04

**Soundness:** 3 good
**Presentation:** 3 good
**Contribution:** 3 good
**Rating:** 5
**Confidence:** 3

**Summary:**

This paper proposes a causal trajectory prediction (CTP) model that aims to provide causal interpretable predictions and estimate treatment effects to enhance the decision-making perspective of machine learning algorithms applied to non-communicable disease management. The progression trajectories of each feature are estimated via ODEs parameterized by feed-forward networks. A constraint is used to ensure that the feed-forward networks can learn a DAG relationship. The paper also explores estimating treatment effect with the learned causal structure.

**Strengths:**

- The target problem is quite significant, especially with the application to non-communicable diseases;
- The proposed idea of using neural ODE and DAG constraint to predict future trajectory is interesting;
- The paper is overall easy to follow.
- Empirical experiments show improvement compared with baseline methods.

**Weaknesses:**

- The proposed method uses one more dimension in $D$ to model the causal relationship between features and hidden confounders; it is unclear how this is done. In [1], a hidden variable is learned to recover hidden confounders, but it is achieved through the independence of multiple causes. In the proposed model, it is unclear how it captures the hidden confounder, and no theoretical insights are given.
- Eq. (6) needs more clarification. It is unclear about the dimensions of $W_1, ..., W_N$, how $C$ is obtained, why it is a vector, etc.
- In Section 2.4, it is assumed "all these parameters are located in a connected region." First, a "connected region" needs to be religiously defined as it can mean differently in different fields (e.g., optimization and graph theory); second, this assumption does not seem obvious. More illustration is expected about when and why this assumption holds.
- When estimating the treatment effect, deconfounding methods are not used, but rather claims that "estimates the feature trajectories and probable bounds under a treatment". The exact meaning of this sentence is not very obvious to me. Since deconfounding has been widely deemed necessary, strong justification for not applying deconfounding would be expected with some grounded theoretical analysis.
- Ablation study is not provided.
- There are no theoretical insights provided about the identifiability and consistency of the developed method.
- Minor type: "There maybe infinite" --> "There may be infinite" in the first paragraph in section 2.4.


[1] Wang, Yixin, and David M. Blei. "The blessings of multiple causes." Journal of the American Statistical Association 114.528 (2019): 1574-1596.

**Questions:**

Please see the Weaknesses part.